# Adenosine leakage from perforin-burst extracellular vesicles inhibits perforin secretion by cytotoxic T-lymphocytes

Hiroko Tadokoro[1], Akiyoshi Hirayama[2], Ryuhei Kudo[2], Masako Hasebe[2], Yusuke Yoshioka[1,3], Juntaro Matsuzaki[1], Yusuke Yamamoto[1], Masahiro Sugimoto[2], Tomoyoshi Soga[2], Takahiro Ochiya[1,3]*

1 Division of Cellular Signaling, National Cancer Center Research Institute, Tokyo, Japan, 2 Institute for Advanced Biosciences, Keio University, Yamagata, Japan, 3 Department of Molecular and Cellular Medicine, Institute of Medical Science, Tokyo Medical University, Tokyo, Japan

* tochiya@tokyo-med.ac.jp

**Data Availability Statement:** All relevant data are within the manuscript and its Supporting Information files.

## Abstract

Extracellular vesicles (EVs) in the tumor microenvironment facilitate intercellular communication. Cancer cell-derived EVs act as an immunosuppressor by transporting cargos and presenting transmembrane proteins. By contrast, CD8+ cytotoxic T-lymphocytes (CTLs) exert anti-cancer cytotoxicity via the pore-forming protein perforin. Here, we hypothesize that although EVs are destroyed by perforin, cancer cell-derived EVs might possess mechanisms that enable them to avoid this destruction. We used a breast cancer cell line, MDA-MB-231-luc-D3H2LN (D3H2LN), to generate EVs. Destruction of the EVs by perforin was demonstrated visually using atomic force microscopy. To investigate immunosuppressive metabolites within cancer cell-derived EVs, we performed metabolomic profiling of EVs from D3H2LN cells cultured for 48 h with or without IFN-γ, which induces metabolic changes in the cells. We found that both types of EV from IFN-γ treated D3H2LN cells and non-treated D3H2LN cells contained adenosine, which has immunosuppressive effects. When we exposed cancer cell-derived EVs to CTLs, perforin secretion by CTLs fell significantly. In addition, the decreases in perforin secretion were ameliorated by treatment with adenosine deaminase, which degrades extracellular adenosine. Taken together, these results suggest that after perforin secreted by CTLs disrupts the membrane of EVs, adenosine released from the EVs acts as an immunosuppressive metabolite by binding to the adenosine receptor on the CTL membrane. This mechanism provides a novel survival strategy using cancer cell-derived EVs.

## Introduction

Extracellular vesicles (EVs) derived from cancer cells are key elements that regulate the tumor microenvironment. Several studies report that EVs derived from cancer cells have immunosuppressive functions [1,2]. Furthermore, some studies report that EVs suppress the function of CD8+ cytotoxic T-lymphocytes (CTLs), which play an important role in anti-tumor immunity [3,4].

**Funding:** This work was supported by the JSPS Early-Career Scientists KAKENHI grant number: JP18K15258 (to H.T.);https://www.jsps.go.jp/english/index.html, the JSPS Grant-in-Aid for Scientific Research on Innovative Areas KAKENHI grant number: JP18H04804 (to A.H.);https://www.jsps.go.jp/english/index.html, the Research on Development of New Drugs GAPFREE (to A.H.) from the Japan Agency for Medical Research and Development (AMED);https://www.amed.go.jp/en/index.html, and Project for Cancer Research and Therapeutic Evolution (P-CREATE) grant number: 19cm0106402h0004 (to T.O.) from the Japan Agency for Medical Research and Development (AMED);https://www.amed.go.jp/en/index.html. The funders had no role in study design, data collection and analysis, decision to publish, or preparation of the manuscript.

**Competing interests:** The authors have declared that no competing interests exist.

Perforin/granzyme B-induced apoptosis is the main cytotoxic pathway induced by CTLs. Perforin forms polymers that insert into the cell membrane to form pores. The underlying mechanism is poorly understood, but it is believed that polymerized perforin forms a cylindrical hydrophobic channel (around 5–20 nm in diameter) that allows free non-selective passive transport across cell membranes [5].

EVs contain various molecules, including proteins, mRNAs, microRNAs, and metabolites. Zhao *et al.* suggest that EVs from cancer-associated fibroblasts contain various metabolites, which they supply to cancer cells [6]; however, the functions of these metabolites remain largely unknown. Thus, we hypothesized that perforin forms pores in the EV membrane, thereby enabling release of the packaged cargos. Here, we focus on ionic metabolites within EVs. To perform ionic metabolome analysis of EVs, we collected EV samples derived from D3H2LN cells in the presence or absence of IFN-γ. Comparison of changes in metabolites in EVs with intracellular metabolic changes makes it possible to detect cell-derived metabolites in EVs; in addition, it excludes contamination from elements in the cell culture medium (CCM).

Under *in vivo* conditions, IFN-γ is secreted by immune cells (mainly natural killer cells and CTLs); it then activates CTLs, which triggers perforin release [7]. In the case of tumors, IFN-γ exposure induces protection from these immune cells. IFN-γ exposure induces expression of many genes to create an immunosuppressive microenvironment. Since some of these genes encode metabolic enzymes, IFN-γ exposure alters cancer metabolism; for example, indoleamine 2,3-dioxygenase 1 (IDO1) induced by IFN-γ is the rate-limiting enzyme of tryptophan catabolism via the kynurenine pathway. Metabolic changes mediated by IDO1 enhance the immunosuppressive properties of the tumor microenvironment [8]. Recent studies show that activation of immune cells is dependent on their metabolism, and that cancer cells use metabolites to control immune cells within the tumor microenvironment [9,10]. In addition, some studies demonstrated that cancer cells exposed to IFN-γ secrete immunosuppressive EVs. Exposure of melanoma cells and glioblastoma cells to IFN-γ increases expression of programmed death-ligand 1 (PD-L1) by their secreted EVs [4,11]. Chen *et al.* clearly showed that PD-L1 in EVs inhibits CD8+ T-cells and facilitates melanoma progression both *in vitro* and *in vivo* [11]. However, the immunosuppressive metabolites contained within EVs exposed to IFN-γ remain unknown.

We hypothesized that exposure to perforin might induce the release of metabolites from EVs. Here, we provide data showing that EVs secreted by breast cancer cells are disrupted by perforin, and that the released metabolites (the amounts of which increased after exposure to IFN-γ) inhibit perforin secretion by CTLs.

## Materials and methods

### Cells and culture conditions

The breast cancer cell line MDA-MB-231-luc-D3H2LN (D3H2LN) (Xenogen Corporation, California, USA) was used as a source of EVs. Human CTLs isolated from human peripheral blood mononuclear cells (PBMCs) (STEMCELL Technologies, Vancouver, Canada) were used as recipient cells for the perforin secretion assay (CTL isolation is described below). All cells were cultured at 37°C in a humidified atmosphere of 5% $CO_2$. D3H2LN cells were maintained in RPMI-1640 medium (Thermo Fisher Scientific, Massachusetts, USA) supplemented with 10% heat-inactivated fetal bovine serum (FBS) (Thermo Fisher Scientific) and Antibiotic-Antimycotic 1× (Thermo Fisher Scientific); this medium was designated "maintenance medium". To generate EVs, D3H2LN cells were cultured in EV medium (advanced RPMI-1640 medium; Thermo Fisher Scientific) supplemented with 2 mM L-glutamine (Thermo Fisher Scientific) and Antibiotic-Antimycotic (1×). In addition, D3H2LN cells were exposed to 50 ng/mL

recombinant human IFN-γ (Peprotech, Inc., New Jersey, USA) diluted in PBS/0.1% BSA (Sigma-Aldrich, Missouri, USA). Control (vehicle)-treated D3H2LN cells were exposed to 0.1% BSA (1 μL/mL). When changing the medium to EV medium, cells were washed twice with PBS to remove FBS. CTLs were isolated from PBMCs using the EasySep Human CD8 + T-Cell Isolation Kit (STEMCELL Technologies). After isolation, CTLs were cultured for 72 h in advanced RPMI-1640 medium supplemented with 2 mM L-glutamine, 10% FBS, Antibiotic-Antimycotic (1×), and rIL-2 (30 U/mL; Peprotech, Inc.) in the presence of Dynabeads Human T-Activator CD3/CD28 (Thermo Fisher Scientific).

## Preparation of the EV fraction

D3H2LN cells were seeded at a density of $3 \times 10^6$ cells/150 mm$^2$ dish and cultured in maintenance medium. After 12 h, the medium was changed to EV medium, and cells were cultured for 48 h. The culture medium was collected and centrifuged at 2000g for 10min at 4˚C, and the supernatant was filtered through a 0.22μm filter (Millipore, Massachusetts, USA). The filtrate was ultra-centrifuged at 110,000g for 70min at 4˚C in a Beckman SW41Ti rotor (Beckman Coulter, Inc., California, USA). The pellets were washed with PBS and ultra-centrifuged again at 110,000g for 70min at 4˚C (Beckman SW41Ti rotor). The number and size of EVs were measured by Nanoparticle Tracking Analysis (NTA) using the Nanosight LM10 system (Nanosight, Amesbury, UK) equipped with a blue laser (405 nm). The protein concentration of EVs was measured using a Micro BCA protein assay kit (Thermo Fisher Scientific). The EVs were characterized according to the MISEV 2018 guidelines [12].

## Phase-contrast transmission electron microscopy

Isolated EVs were visualized by Terabase Inc. (Aichi, Japan) using a phase-contrast transmission electron microscope, which generates high-contrast images of the nanostructures comprising soft materials (including biological samples such as liposomes, viruses, bacteria, and cells) without the need for staining processes that may damage samples. The natural structure of a sample distributed in solution can be observed by preparing the sample using a rapid vitreous ice-embedding method, followed by cryo-phase-contrast transmission electron microscopy.

## Immunoblot analysis

Cells and EVs were lysed in M-PER Mammalian Protein Extraction Reagent (Thermo Fisher Scientific). Samples were prepared in 4× Laemmli Sample Buffer (Bio-Rad Laboratories, California, USA) containing 2-mercaptoethanol (Bio-Rad Laboratories). Equal amounts of protein were separated on SDS-PAGE gels using a Mini-PROTEAN® Tetra Vertical Electrophoresis Cell (#1658004JA, Bio-Rad Laboratories) and a PowerPac™ HC power supply (#1645052, Bio-Rad Laboratories). Next, the proteins were transferred to a PVDF membrane (#1620174, Bio-Rad Laboratories) by wet transfer (Mini Trans-Blot® Module, #1703935JA; Bio-Rad Laboratories). Membranes were blocked using Blocking One (NACALAI TESQUE, INC., Kyoto, Japan). Antibodies specific for indoleamine 2,3-dioxygenase 1 (IDO1) (NBP1-87702; Novus Biologicals, Colorado, USA), β-actin (AC-15; Santa Cruz Biotechnology Inc., California, USA), PD-L1 (13684S; Cell Signaling Technology Inc., Massachusetts, USA), CD9 (12A12; Cosmo Bio Co., Ltd., Tokyo, Japan), CD63 (8A12; Cosmo Bio Co., Ltd.), Alix (ABC40; Millipore), Flotillin-1 (610820; BD Biosciences, New Jersey, USA), TSG101 (612696; BD Biosciences), and Apolipoprotein A I (APOA1) (AF3664; R & D Systems, Minnesota, USA) were used as primary antibodies. HRP-labeled anti-rabbit IgG antibody (GE Healthcare Japan Co., Ltd., Tokyo, Japan), anti-mouse IgG antibody (GE Healthcare Japan Co.), and Peroxidase

AffiniPure Donkey Anti-Goat IgG antibody (705-035-147; Jackson ImmunoResearch Inc., Pennsylvania, USA) were used as secondary antibodies. The chemiluminescence reagent used to visualize β-actin and Alix was Western Lightning Plus-ECL (PerkinElmer, Massachusetts, USA). ImmunoStar LD (FUJIFILM Wako Pure Chemical Corporation, Osaka, Japan) was used to visualize other proteins. Luminescent images were analyzed with a Lumino Imager (LAS-3000; Fujifilm, Tokyo, Japan).

## High-speed atomic force microscopy (HS-AFM)

HS-AFM (in AC mode in liquid) was performed using SS-NEX (Research Institute of Biomolecule Metrology Co., Ltd., Ibaraki, Japan). Two kinds of cantilever, BL-AC10DS-A2 (Olympus Corporation, Tokyo, Japan) and USC-F1.2-k0.15 (NanoWorld, Neuchatel, Switzerland), were used. The scan areas measured $200 \times 200$ pixels. All observations were performed at room temperature (21˚C). A droplet of EV solution (14–18 ng in PBS) was incubated for 10 min on a mica surface in a wet environment. Next, the mica was washed two or three times with 10 μL PBS. The sample was then placed in a liquid cell containing almost 150 μL of PBS and analyzed by HS-AFM. During HS-AFM, 2 μL of 0.1 M $CaCl_2$ was injected into the liquid cell, followed by 1 μL of perforin (20 ng/μL). This experiment was performed once for each EV.

## Measurement of the EV destruction by perforin

Perforin was added to EVs (containing 6 μg protein) in $Ca^{2+}$-HEPES buffer, according to the manufacturer's instructions (Enzo Life Sciences Inc., New York, USA). The mixture was incubated at 37˚C for 15 min. Next, the samples were divided in half, and RNase A (Qiagen, Hilden, Germany) was added to one of the two. After addition of RNase A, the samples were incubated at 37˚C for 30 min. To isolate RNA, 700 μL QIAzol lysis reagent (Qiagen) was added to each sample, followed 5 min later by 10 μL of 10 nM miRNeasy Serum/Plasma Spike-In Control (cel-miR-39; Qiagen) (S4 Fig). Isolation of miRNAs from EVs was performed using a miR-Neasy kit (Qiagen). Levels of miR-16 and cel-miR-39 were measured in TaqMan miRNA assays using real-time PCR. All TaqMan microRNA assays were purchased from Applied Biosystems (Massachusetts, USA). Real-time PCR was carried out using a StepOne Real-Time PCR System (Applied Biosystems), according to the manufacturer's recommended program.

## Sample collection and metabolite extraction

To prepare the EV fraction, D3H2LN cells were seeded and cultured as described above (S5 Fig). After the supernatant was collected, cells were obtained by trypsinization. Next, cells ($5 \times 10^6$) were collected, washed with PBS, and centrifuged at 2000g for 5min at 4˚C. After washing twice, 1 mL methanol containing internal standards (20 μmol/L each of methionine sulfone and camphor 10-sulfonic acid) was added. Metabolites were extracted from 400 μL lysate by addition of 160 μL water and 400 μL chloroform. Next, 300 μL aqueous phase was passed through a 5 kDa cutoff centrifuged filter tube (Human Metabolome Technologies, Tsuruoka, Japan). The filtrate was dried in a vacuum centrifuge and resuspended in 50 μL water prior to metabolome analysis. Next, 264 mL cell supernatant was ultra-centrifuged at 110,000g for 70min at 4˚C in a Beckman SW41Ti rotor. The supernatant was collected and used as a sample of CCM. Next, 100 μL CCM was mixed with 400 μL methanol containing internal standards (20 μmol/L each of methionine sulfone and camphor 10-sulfonic acid), 60 μL water, and 400 μL chloroform. The extraction steps were identical to those described for the cell samples. The pellets were washed with saline and ultra-centrifuged twice at 110,000g for 70min at 4˚C in a Beckman SW41Ti rotor. After washing, the pellets were diluted to a final volume of 50 μL in saline. Then, 6 μL pellet solution was added to 19 μL saline; this solution

was used for normalization. The remaining pellet solution (44 μL) was used as the EV sample; this was mixed with 200 μL methanol containing internal standards (40 μmol/L methionine sulfone and 1 μmol/L camphor 10-sulfonic acid), and 50 μL chloroform. Finally, 140 μL of the upper aqueous layer was transferred to a glass tube and dried in a vacuum centrifuge. Dried samples were dissolved in 20 μL water immediately prior to metabolome analysis.

## Metabolome analysis

Cell and CCM samples were analyzed by capillary electrophoresis-mass spectrometry (CE-MS). CE-MS-based metabolome analysis and data analysis were performed essentially as described [13–15]. Anionic metabolites in EVs were analyzed by capillary ion chromatography-mass spectrometry (IC-MS), as described previously [16]. Analysis of cationic metabolites in EVs was performed using liquid chromatography-mass spectrometry (LC-MS). LC-MS analysis was performed using an Agilent 1290 Infinity LC system equipped with a Q Exactive Orbitrap MS system (Thermo Fisher Scientific). Separations were performed on a Develosil RPAQUEOUS-AR-3 column (2 × 250 mm; Nomura Chemical Co., Ltd., Aichi, Japan) maintained at 30°C. The mobile phase consisted of 10 mM acetic acid (solution A) and methanol (solution B). The gradient of solution B was as follows: 0% from 0 min to 7 min, 50% at 30 min, and 99% at 30.1 min; this was maintained until 35 min. The flow rate was 0.2 mL/min, and the injection volume was 1 μL.

The Q Exactive mass spectrometer was operated in ESI positive ion mode, and the spray voltage was set at 3.5 kV. The capillary temperature was 350°C, the sheath gas flow rate was 40, the auxiliary gas flow rate was 10, the sweep gas flow rate was 0, and the S-lens was 35 (arbitrary units). The parameters for the full MS scan were as follows: resolution, 70,000; auto gain control target, $3 \times 106$; maximum ion injection time, 100 ms; and scan range, 70–1050 m/z. The instrument was calibrated at the beginning of each sequence using the calibration solution provided by the instrument manufacturer. The raw data obtained by LC-MS were analyzed using TraceFinder software (version 3.2, Thermo Fisher Scientific).

## Perforin secretion assay

CTLs were collected after 72 h of culture with Dynabeads (Thermo Fisher Scientific), which were removed according to the manufacturer's instructions. CTLs were seeded at a density of $1.5 \times 10^5$–$2 \times 10^5$ cells/well and treated with EVs (4 μg/mL) from vehicle-treated D3H2LN cells (vehicle EVs), EVs (4 μg/mL) from IFN-γ-treated D3H2LN cells (IFN-γ EVs), or 10 μM adenosine (Sigma-Aldrich). To decompose the adenosine, 1 U adenosine deaminase (ADA: Sigma-Aldrich) was added to each well. Perforin protein levels in the supernatant were measured using a Human Perforin ELISA kit (Abcam, Cambridge, UK).

## Statistical analysis

Data are presented as the mean ± S.D. of three biological samples analyzed in triplicate. Two treatment groups were compared using Student's *t*-test. Multiple group comparisons were performed using the Tukey–Kramer method after one-way analysis of variance. Results were considered statistically significant at $P < 0.05$.

## Results

### Perforin bursts EVs

Although it is known that perforin polymerizes and forms pores in cell membranes, the effect of perforin on EVs is unknown. To investigate this, we used AFM to visualize EVs after

exposure to perforin (Fig 1, S1–S3 Movies, and S1–S3 3D Movies). As a negative control, CaCl$_2$ was added to EVs, and any changes were observed (S1 Movie and S1 3D Movie). EVs were isolated from the culture medium of D3H2LN cells grown in the presence or absence of IFN-γ (which increases metabolic activity without affecting cell growth and or the size of EVs) (S1 Fig) and subjected to metabolome analysis to identify metabolites within the EVs. In both EVs from vehicle-treated D3H2LN cells (vehicle EVs) and EVs from IFN-γ-treated D3H2LN cells (IFN-γ EVs), addition of perforin caused some EVs to burst; indeed, the membrane structures were clearly destroyed (Fig 1 and S2 Fig, S2–S5 Movies, and S2–S5 3D Movies). This suggests that the packaged cargo could be released from the EVs. Furthermore, we observed that some IFN-γ EVs had shrunk prior to bursting (S2 Fig, S5 Movie, and S5 3D Movie).

Cross-sectional image analysis of each photo (S1 Movie, Fig 1, and S2 Fig) was performed to clarify the size of each EV. From this, diameters were calculated (S3 Fig). We found that the measurements were not significantly different between vehicle EVs and IFN-γ EVs; neither was there a difference in EV size distribution, as measured by NTA (S1F Fig).

## Measurement of the EVs destruction by perforin

Next, we measured perforin-mediated EV membrane destruction of vehicle EVs and IFN-γ EVs by real-time PCR (S4 Fig). EV membrane destruction was assessed by measuring the amount of miR-16 in EVs before and after RNase A treatment. In this analysis, the amount of miR-16 was used simply to indicate the integrity of the EV membrane; if the EV membrane is damaged, then RNase A has access to RNA molecules inside the EVs and can degrade them. In the absence of perforin, RNase A did not affect the amount of intact miR-16 (compared with that in control samples) (Fig 2A, left panel and Fig 2B, left panel). By contrast, perforin (200 ng/mL) plus RNase A treatment led to a clear reduction in the amount of miR-16 in both IFN-γ EVs and vehicle EVs (Fig 2A, right panel and Fig 2B, right panel). These results suggest that perforin disrupts the EV membrane.

## IFN-γ alters D3H2LN cell metabolism

A number of studies show that IFN-γ promotes tumor survival and progression [17,18]. One role of IFN-γ is regulation of the immunosuppressive tumor microenvironment [19]. IFN-γ induces expression of genes encoding IDO1 and PD-L1, which are associated with cancer cell immune evasion [3,20]. As a first step to identify immunosuppressive metabolites, D3H2LN cells were treated with IFN-γ (S1C and S1D Fig). To confirm the effect of IFN-γ on cell metabolism, we performed metabolome analysis on vehicle cells and IFN-γ cells using LC-MS.

The amounts of kynurenine and phosphoenolpyruvate (PEP) in IFN-γ cells were significantly higher than those in vehicle cells. By contrast, the amounts of tryptophan, citrate, cis-aconitate, succinate, fumarate, and malate were significantly lower in IFN-γ cells than in vehicle cells. IFN-γ increased glycolysis and decreased the TCA cycle in the cells (Fig 3).

## Metabolomic profiling of IFN-γ EVs

As expected, IFN-γ treatment promoted catabolism of tryptophan in cells; however, the rate-limiting enzyme of tryptophan catabolism, IDO1, was not detected in EVs (S1E Fig). IFN-γ treatment did not affect the size of EVs (S1F Fig). By contrast, PD-L1 protein levels increased in IFN-γ EVs (S1E Fig), as reported previously [4,11].

Vehicle EVs and IFN-γ EVs (collected according to the scheme shown in S5 Fig), were subjected to metabolome analysis using LC-MS. When considering the effects of EV metabolites on cells, an important factor is the absolute amount of EV metabolites. First, metabolites in EVs were normalized according to the amount of EV protein. Metabolites in EVs, which

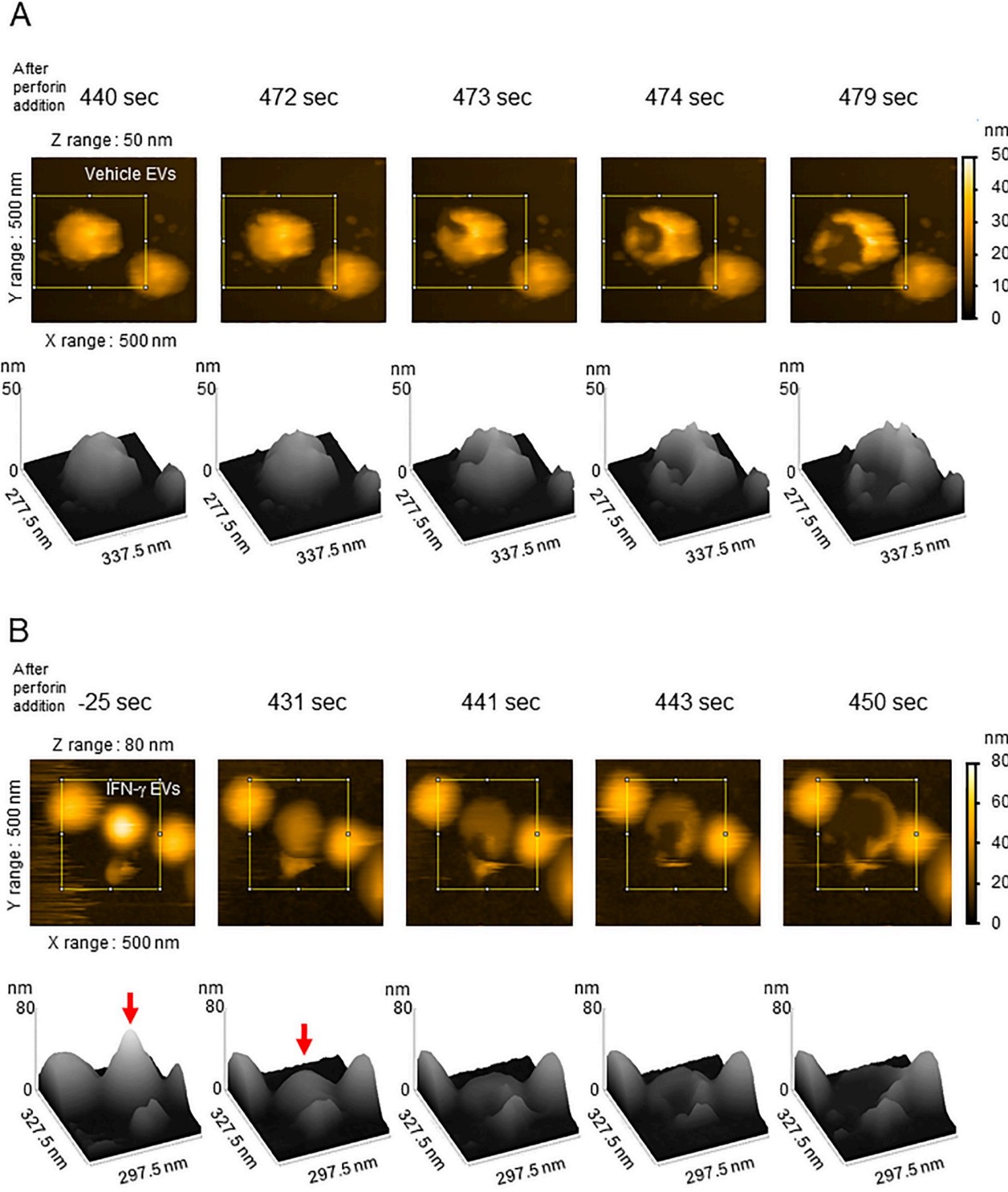

**Fig 1. Successive AFM images showing perforin-mediated disruption of EV membranes.** (A) A solution containing EVs from vehicle-treated D3H2LN cells (vehicle EVs) (18 ng in PBS) was incubated for 10 min on a mica surface in a hsumidified environment. The mica was then washed three times with PBS and imaged under an atomic force microscope (S2 Movie and S2 3D Movie). BL-AC10DS-A2 was ued as a cantilever. (B) Some EVs from IFN-γ-treated D3H2LN cells (IFN-γ EVs) burst after shrinking in response to perforin (S3 Movie and S3 3D Movie). The 3D images (processed using ImageJ software) derived from successive AFM images show a height reduction in IFN-γ EVs (red arrow).

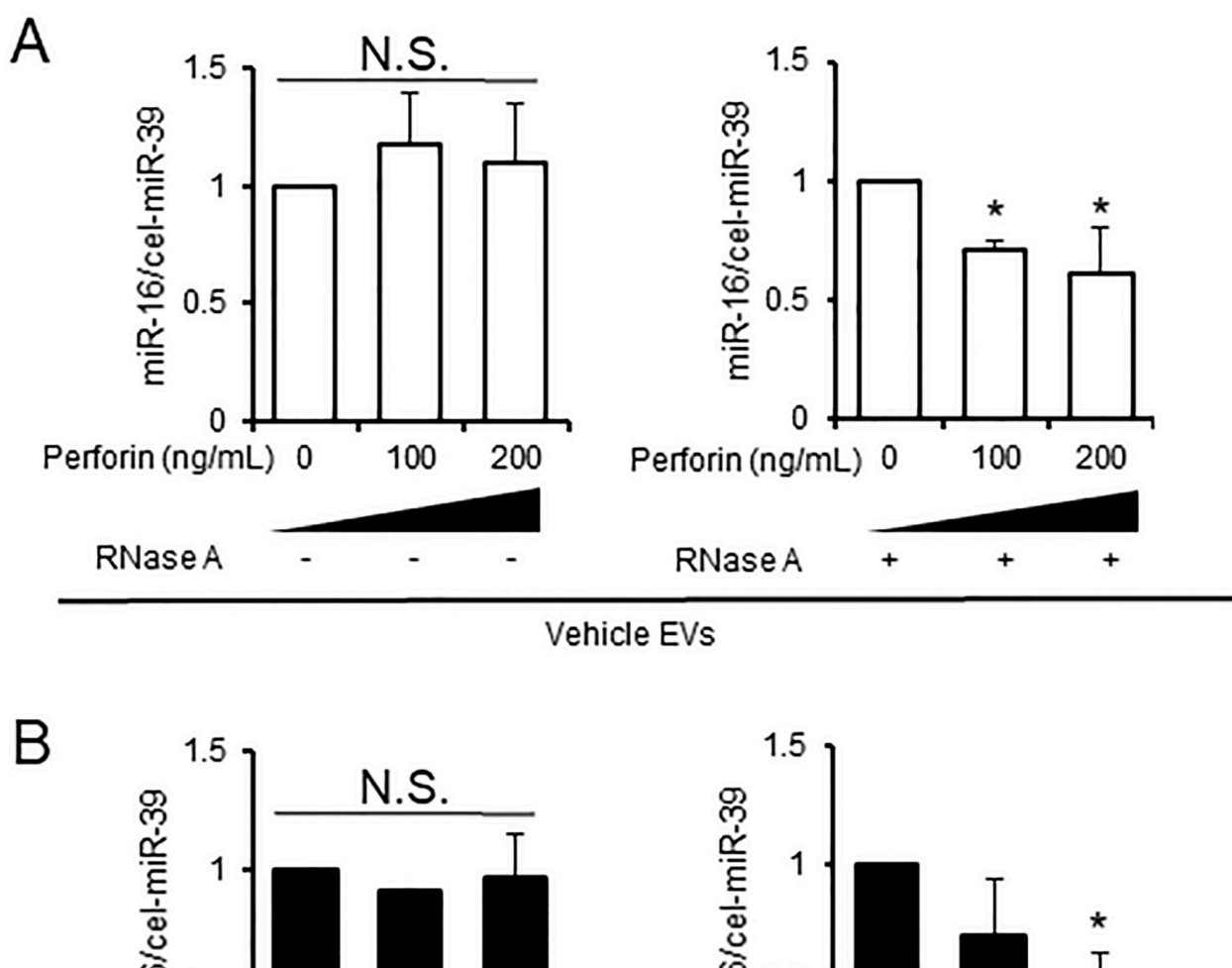

**Fig 2. Measurement of the EV destruction by perforin.** (A) EVs from vehicle-treated D3H2LN cells (vehicle EVs). (B) EVs from IFN-γ-treated D3H2LN cells (IFN-γ EVs). Both types of EV were treated with or without perforin, and miRNAs were degraded by RNase A (right panels). MiRNA degradation was measured by real-time PCR; cel-miR-39 was used as a spike-in control. $P < 0.05$ (Tukey–Kramer method). All experiments were performed using three biological replicates, and each sample was compared with individual controls; thus the control has no standard deviation.

account for 3% of total metabolites mole in each EV, were then ranked according to the metabolome analysis results. The top four metabolites (UDP-N-acetylglucosamine, uracil, uridine, and adenosine) in IFN-γ EVs were identical to those in vehicle EVs (Table 1). The concentrations of most metabolites derived from vehicle EVs and IFN-γ EVs were very low, accounting

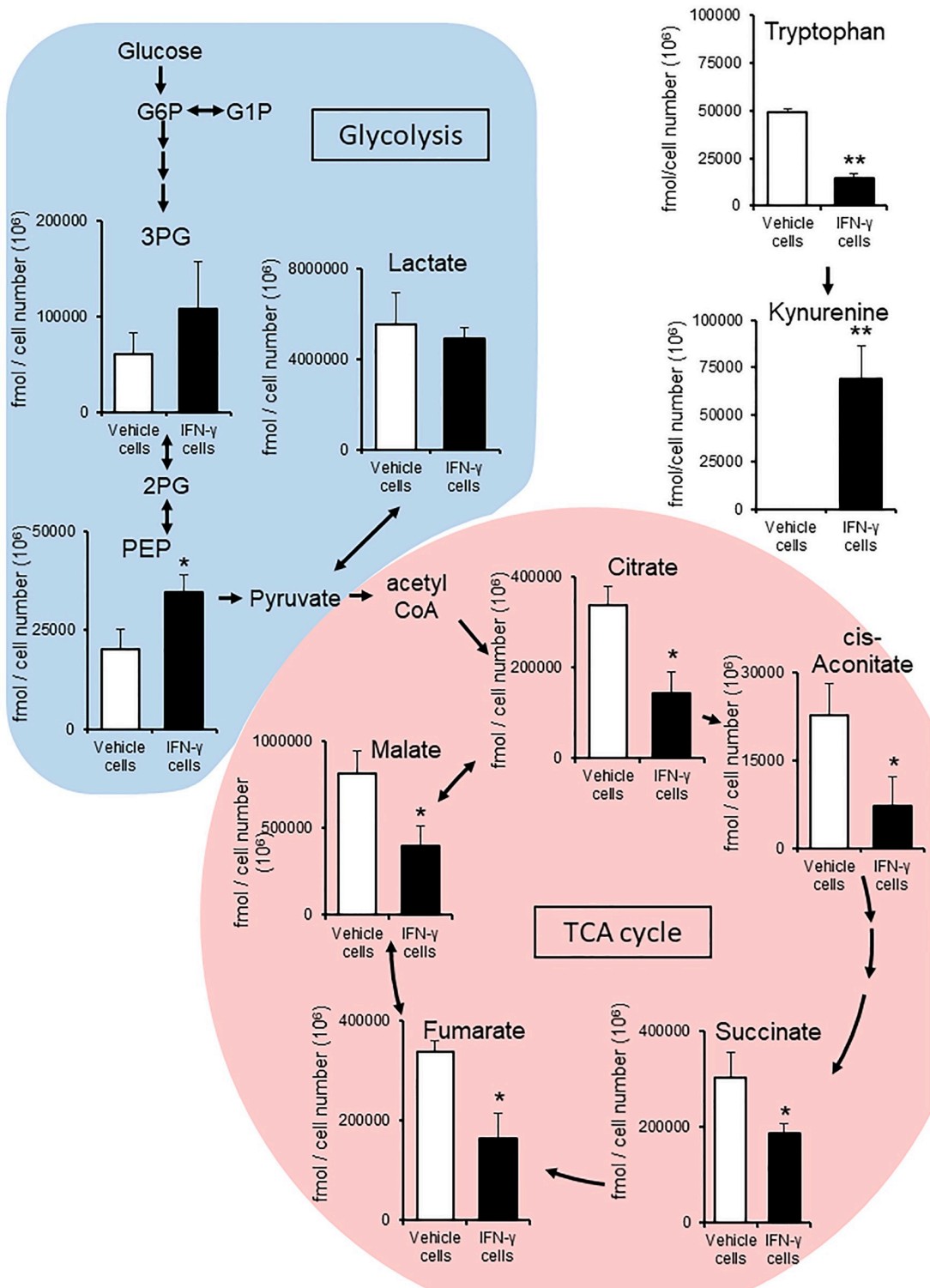

**Fig 3. Metabolome data map of metabolites involved in the glycolysis, TCA cycle, and tryptophan catabolism pathways, as detected in D3H2LN cells treated with or without IFN-γ.** (A) After IFN-γ treatment, there was a significant difference in the amount of components related to glycolysis and the TCA cycle; PEP ($P = 0.041$), citrate ($P = 0.011$), cis-aconitate ($P = 0.040$), succinate ($P = 0.042$), fumarate ($P = 0.011$), and malate ($P = 0.028$). (B) The amounts of tryptophan ($P = 6.4E^{-5}$) and kynurenine ($P = 0.010$), components of tryptophan catabolism. Kynurenine increased significantly in the presence of IFN-γ. Columns, average concentration (fmol/cell number ($10^6$)); bars, S.D. (n = 3). $^*P < 0.05$, $^{**}P < 0.01$ (Student's $t$-test). PEP, phosphoenolpyruvate. All experiments were performed using three biological replicates.

**Table 1. Ranking of ionic metabolites in D3H2LN cell-derived EVs.**

| Rank | Vehicle EVs | (%) | IFN-γ EVs | (%) |
|---|---|---|---|---|
| 1 | UDP-N-acetylglucosamine | 18.7 | UDP-N-acetylglucosamine | 15.6 |
| 2 | Uracil | 10.2 | Uracil | 13.3 |
| 3 | Uridine | 8.8 | Uridine | 12.3 |
| 4 | Adenosine | 7.8 | Adenosine | 9.6 |
| 5 | ADP | 4.5 | Guanosine | 3.7 |
| 6 | Glycerophosphorylcholine | 4.2 | G1P | 3.5 |
| 7 | Ethanolamine phosphate | 3.6 | Glycerophosphorylcholine | 3.4 |
| 8 | G1P | 3.4 | | |
| 9 | UMP | 3.1 | | |
| 10 | GDP | 3 | | |

Metabolites in EVs from vehicle-treated D3H2LN cells (vehicle EVs) and EVs from IFN-γ-treated D3H2LN cells (IFN-γ EVs), which account for 3% of total metabolites in each EV, are ranked according to the amount detected. The percentage of each metabolite in both types of EVs cannot be compared directly because the average amount of total metabolites in IFN-γ EVs was 2.4 μM, whereas that in vehicle EVs was 1.9 μM. All experiments were performed using three biological replicates. UDP-N-acetylglucosamine, uridine diphosphate N-acetylglucosamine; ADP, adenosine diphosphate; G1P, glucose-1-phosphate; UMP, uridine monophosphate; GDP, guanosine diphosphate.

for only 0–0.9% of total EV metabolites (S1 Table). The absolute value of the contribution rates (more than 3.0) between IFN-γ EVs and vehicle EVs was also ranked (Table 2). The ratio of the quantitative change of an individual metabolite in EVs caused by IFN-γ to that of all metabolites in EVs caused by IFN-γ is expressed as a contribution rate. This was calculated using the formula described in the legend to Table 2. In addition, we also analyzed metabolomic profiles according to an analytical scheme to select functional metabolites (S6 Fig). The analysis scheme was designed to remove the effects of contaminants in the culture medium. Several metabolites (uracil, uridine, adenosine, guanosine, and inosine) that had contribution rates of > 3% were also identified by the analytical scheme. The amount of guanosine and inosine in vehicle EVs and IFN-γ EVs was less than half that of uracil, uridine, and adenosine (Fig

**Table 2. Top ten absolute values of contribution rates to changes in the amount of ionic metabolites in EVs after IFN-γ exposure.**

| Rank | Metabolites | Contribution rate \|%\| |
|---|---|---|
| 1 | **Uridine** | 24.2 |
| 2 | **Uracil** | 24.1 |
| 3 | **Adenosine** | 15.7 |
| 4 | **Guanosine** | 6.6 |
| 5 | **UDP-glucose** | 5.3 |
| 6 | **UDP-N-acetylglucosamine** | 4.8 |
| 7 | **G1P** | 4 |
| 8 | **N-Acetylglucosamine 1-phosphate** | 3 |
| 9 | ADP | 2.5 |
| 10 | **Inosine** | 2.1 |

The contribution rate represents the ratio of the quantitative change of an individual metabolite in EVs caused by IFN-γ to the quantitative change of all metabolites in EVs caused by IFN-γ. Contribution rates were based on values from metabolome analysis and calculated using the following equation: contribution rate (%) = (the amount of metabolite in IFN-γ EVs—the amount of metabolite in vehicle EVs) / (total amount of metabolites in IFN-γ EVs—total amount of metabolites in vehicle EVs) × 100. Metabolites highlighted in bold were more contained in IFN-γ EVs than in vehicle EVs. UDP-glucose, uridine diphosphate glucose; UDP-N-acetylglucosamine, uridine diphosphate N-acetylglucosamine; G1P, glucose-1-phosphate; ADP, adenosine diphosphate.

4 and S7 Fig). Therefore, it is highly likely that uracil, uridine, and adenosine are functional metabolites in IFN-γ EVs.

The amount of uridine in IFN-γ -treated D3H2LN cells was significantly lower than that in vehicle-treated D3H2LN cells (*$P < 0.05$), but the amount of uridine in IFN-γ EVs was higher than that in vehicle EVs (not significant). The amount of uracil, uridine, adenosine, guanosine, and inosine in CCM could not be measured by CE-MS since the concentrations were too low (Fig 4B and S7 Fig, middle panels).

At first, we expected that kynurenine would be contained in IFN-γ EVs and act as an immunosuppressor. The concentration of kynurenine in IFN-γ EVs was 1.71, 18.25, and 8.90 nM (9.62 ± 6.77 nM) (Table 3). However, the concentration of kynurenine in IFN-γ CCM was 1.60, 1.57, and 1.84 μM (1.67 ± 0.12 μM). The concentration of kynurenine in IFN-γ EVs is much lower than that in IFN-γ CCM. This means that the concentration of kynurenine in IFN-γ EVs was affected by contaminants in the CCM. Therefore, we excluded kynurenine from the analysis of EV metabolites. IFN-γ treatment promoted intracellular tryptophan catabolism and glycolysis; however, these differences were not apparent in EV metabolites between vehicle and IFN-γ treatment. In addition, the difference between metabolites in vehicle EVs and those in IFN-γ EVs were not observed. Therefore, we focused our analysis on vehicle EVs.

## EV-mediated reductions in perforin secretion by CTLs are ameliorated by ADA

Since adenosine has immunosuppressive functions [21,22], we focused on this metabolite. The immunosuppressive function of adenosine is mediated by adenosine receptors [21,23]. In addition, adenosine inhibits perforin expression by activated T-cells [23,24]. To investigate the effect of adenosine released from EVs on perforin secretion by activated T-cells, we exposed CTLs to vehicle EVs or adenosine (Fig 5A). Compared with non-treated CTLs (control), adenosine-treated CTLs showed a significant reduction in perforin secretion ($P < 0.05$). Moreover,

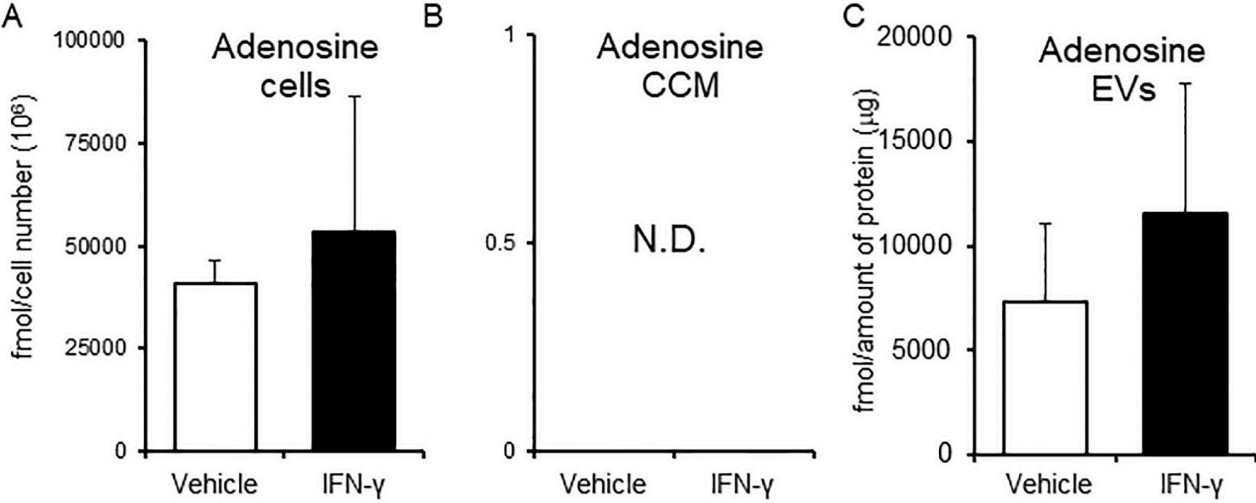

**Fig 4. Adenosine concentration in cells, cell culture medium (CCM), and EVs.** (A)–(C) Adenosine concentrations. D3H2LN cells were incubated in EV medium with or without IFN-γ. After 48 h, cells and supernatant were collected. The supernatant was ultra-centrifuged to separate EVs from the CCM. For metabolome analysis, cells and CCM samples were subjected to CE-MS, and EV samples were subjected to IC-MS and LC-MS. The amount of metabolites was normalized to the amount of protein. Columns, average concentration ((A), fmol/cell number (10^6); (C), fmol/mg protein). Bars, S. D. (n = 3). N.D., the metabolite concentration was below the detection limit of the assay. All experiments were performed using three biological replicates.

**Table 3. Three data sets: IFN-γ EV kynurenine content, IFN-γ CCM kynurenine content, and the IFN-γ EV particle count.**

|  | 1st | 2nd | 3rd |
|---|---|---|---|
| The kynurenine amount of IFN-γ CCM (nM) | 1600 | 1570 | 1840 |
| The kynurenine amount of IFN-γ EVs (nM) | 1.71 | 18.25 | 8.9 |
| The particle number of IFN-γ EVs (E10) | 1.39 | 1.53 | 1.84 |

Metabolome analysis was performed using three biological replicates. Each EV was isolated from the same sample taken from each cell culture medium (i.e., 1st EVs were isolated from the 1st cell culture medium (CCM)).

vehicle EVs also caused a significant reduction in perforin secretion by CTLs ($P < 0.05$). Also, the EV-mediated reduction in perforin secretion by CTLs was ameliorated by adenosine deaminase (ADA) (Fig 5B). These results suggest that EV-mediated reduction in perforin secretion by CTLs is caused by extracellular adenosine released by vehicle EVs after exposure to perforin.

## Discussion

Cancer cells secrete EVs, which regulate the tumor microenvironment and enable survival and proliferation [25–27]. Some EVs secreted by cancer cells suppress the activity of CTLs. IFN-γ is one of the major cytokines that activates CTLs to trigger anti-tumor immunity [28–30]. However, several cancers acquire immune resistance by expressing immune-evasive gene signatures in response to IFN-γ, thereby creating a "comfortable" tumor microenvironment for survival [3,31]. The pore-forming protein perforin is one of the major cytotoxic mediators

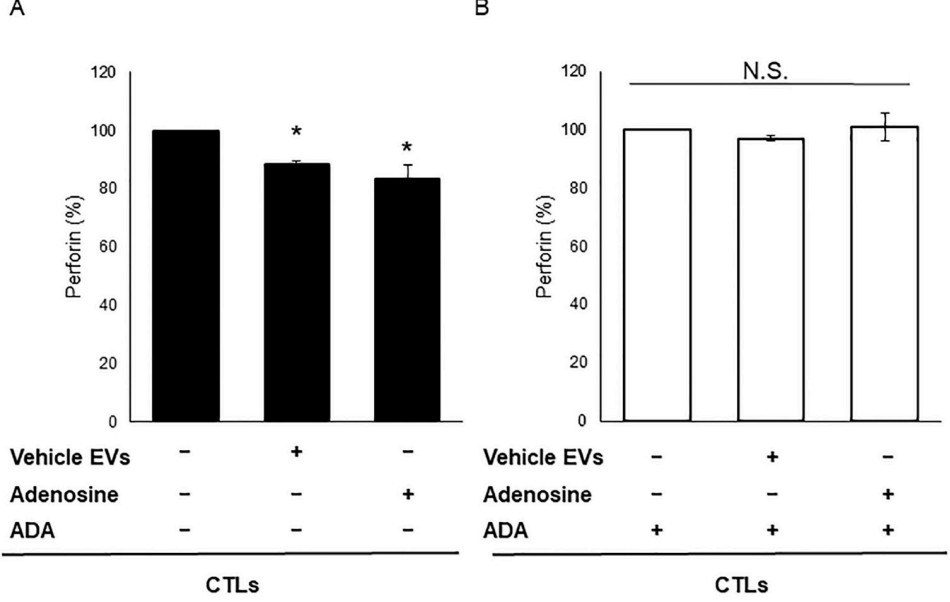

**Fig 5. Effect of EVs containing adenosine on perforin secretion by CTLs.** Perforin protein levels in medium from CTLs were measured by ELISA after 24 h of stimulation with adenosine or EVs from vehicle-treated D3H2LN cells (vehicle EVs). (A) Levels in the absence of adenosine deaminase (ADA). (B) Levels in the presence of ADA. Data represent the mean ± S.D. (n = 3). $P < 0.05$ (Tukey–Kramer test). All experiments were performed using three biological replicates, and each sample was compared with its individual control; therefore, the control has no standard deviation.

secreted by activated CTLs; this secretion is inhibited by adenosine via adenosine receptors. Here, we show that perforin from CTLs disrupts EVs secreted by tumor cells; however, adenosine released by disrupted EVs acts as an immunosuppressor, which reduces perforin secretion by CTLs (Fig 6).

The AFM data clearly demonstrate perforin-mediated disruption of EV membranes. These results suggest that perforin disrupts EVs via two mechanisms: membrane bursting and structural shrinkage. We speculate that these differences are related to the pore size of polymerized perforin. Although perforin is thought to form pores of uniform size within artificial lipid layers [32,33], Lic *et al.* reported that perforin forms pores of different diameters (long length: 248 ± 19 nm; short length: 202 ± 11 nm) in OVA-B16 cell membranes [34]. It is also plausible that perforin forms pores of different sizes in cells and EV membranes. We assume that the pore size is controlled by the structural components of the lipid membrane and the complexity of the cell membrane structure. Here, we observed pre-rupture shrinkage only in IFN-γ EVs;

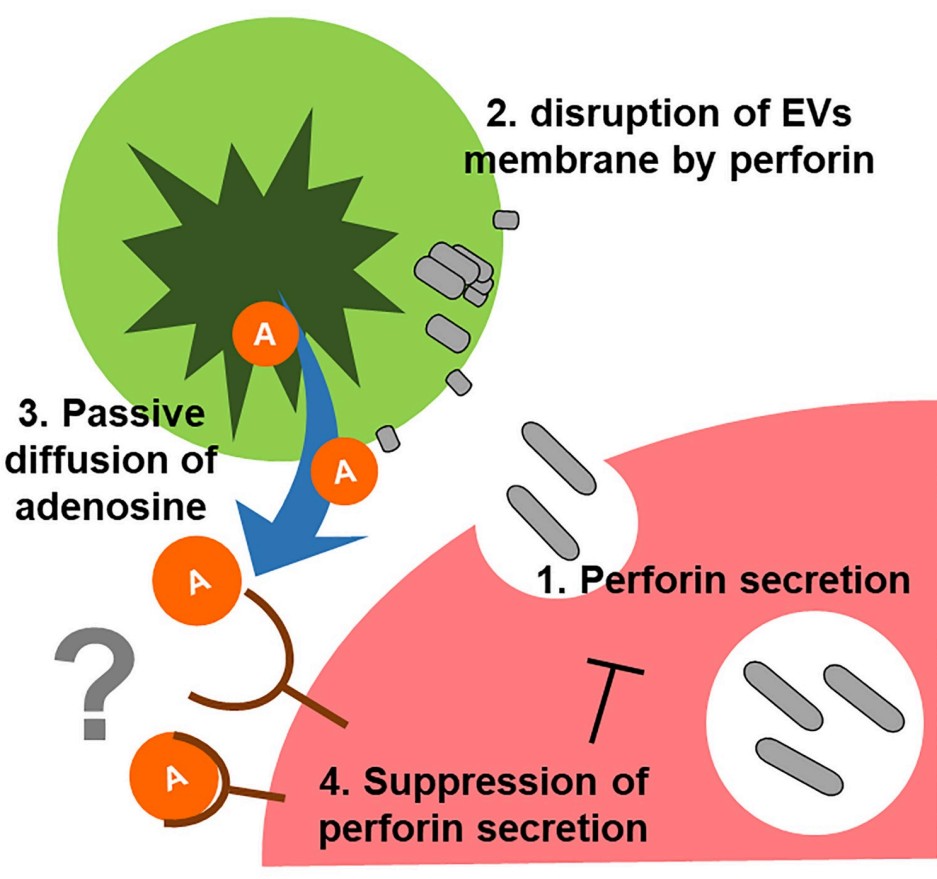

**Fig 6. Schematic illustration of the relationship between perforin-mediated EV destruction and adenosine release from EVs into the tumor microenvironment.** EVs contain adenosine (Tables 1 and 2, and Fig 4C). Adenosine from EVs suppresses immune responses as follows: 1) perforin is secreted by CTLs; 2) perforin inserts into and destroys the EV membrane (Figs 1 and 2); 3) adenosine diffuses passively out of EVs (Fig 2); 4) adenosine suppresses perforin secretion by CTLs (Fig 5A). Since this study did not demonstrate involvement of adenosine receptors in reducing perforin secretion from CTLs, the contribution of adenosine receptors is denoted by a question mark. However, several reports show that adenosine A3 receptors contribute to inhibition of perforin secretion by mouse T-cells [23,24].

however, further experiments are needed to determine the effects of IFN-γ treatment on EV membrane disruption (e.g., shrinkage and the kinetics of disruption). Depending on the setting of AFM, EVs were destroyed and the rate of EVs destruction could not be statistically processed in this study. The kinetics of EV contraction and perforin destruction must be investigated in future, by observing a wider area at AFM-level super-resolution.

Although we focused only on perforin, complement component C9 (a member of the membrane-bound complex) may also disrupt EVs because it shares structural homology with perforin [35]. Furthermore, protein S, an inhibitor of the lytic activity of complement components, may protect EVs from both perforin and C9.

We treated cells with IFN-γ for two reasons. One was to ensure detection of metabolomic profiles by focusing on metabolites associated with changes in intracellular metabolism. The other was to identify metabolites in EVs that are associated with immunosuppression. However, we found that IFN-γ treatment did not affect the metabolic profile in EV metabolites as much as cellular metabolites.

Extracellular adenosine exerts immunosuppressive effects on immune cells via the adenosine receptor [36,37]. Extracellular adenosine is either secreted by cells or generated from extracellular ATP. Extracellular adenosine has a very short half-life (10 s) due to rapid uptake by cells (via nucleoside transporters) and irreversible conversion to inosine by ADA [38–40]. The present study demonstrates that EVs from D3H2LN cells contain adenosine (Fig 4C). Puhka *et al.* detected adenosine in platelet-derived EVs and urinary EVs from prostate cancer patients; adenosine was enriched in each type of EVs [41]. The EVs may package adenosine because the cytosol contains adenosine. However, few reports have examined the role of adenosine in EVs. Adenosine in EVs may be protected from nucleoside transporter-mediated uptake and enzyme-mediated degradation; it also acts as an immunosuppressive metabolite.

Fig 5 shows the effect of extracellular adenosine released from EVs. There are other two possible mechanisms that explain increasing levels of extracellular adenosine. One is transmission via equilibrative nucleoside transporters (ENTs). However, the adenosine concentration in EVs is lower than that required by ENTs. Another is production of extracellular ATP by membrane-bound CD39 and CD73. The effect of CD73 on extracellular adenosine concentrations is one limitation of this study. CD73 is an ecto-5'-nucleotidase that converts extracellular AMP to adenosine [42]. Several studies report that cancer cell-derived EVs express CD73, and that this converts AMP to adenosine [43,44]. Clayton *et al.* reported that extracellular adenosine was generated from extracellular ATP or 5' AMP by CD39 and CD73 expressed on EVs [43]. In their study, they added ATP or 5' AMP to the medium. We did not include ATP or 5' AMP in the advanced RPMI-1640 medium used as the EV medium. Therefore, we do not think that production of adenosine from extracellular ATP via CD39 and CD73 occurred in our experiments. However, we were unable to clarify the effect of CD73 on extracellular adenosine concentrations. Although extracellular adenosine is produced by CD73, EVs also secrete adenosine; therefore, they are at least partially responsible for extracellular adenosine levels.

There are three other limitations to this study. The first is the sensitivity of real-time PCR for miR-16 in EVs. Since miR-16 is used as a housekeeping miRNA in cells, it was used as a marker for membrane disruption due to its high abundance in EVs. However, the Ct value of miR-16 was low, between 29 and 35. For analysis, we used results for which the difference in duplicate Ct values was less than 0.6. The second limitation pertains to intra-EV sample variability and analytic limitations. Metabolome analysis of EV samples is difficult due to contamination by metabolites from the CCM. The effect of contamination depends on the amount of CCM remaining in the EV samples. In addition, we cannot predict the amount of contaminants because the metabolites in CCM samples are too dilute; by contrast, metabolites in EV

samples were concentrated by ultracentrifugation. The third limitation is that the results are not generalizable because they are based on only one cell line.

Taken together, the results presented herein demonstrate that perforin destroys EVs and releases adenosine, which then suppresses perforin secretion by CTLs. Since adenosine signaling is an important pathway that regulates tumor immunity, it is a candidate immunity checkpoint target [21,45]. Therefore, it would be appropriate to examine whether adenosine released from EVs has effects on the tumor microenvironment in addition to those of extracellular free adenosine. Fig 5A shows that perforin secretion from CTLs fell by only 15%. In this study, we used the Dynabeads Human T-Activator CD3/CD28 and IL-2 to activate CTLs; however, specific activation of T-cells recognizing a particular antigen, like ovalbumin, may have been more biologically significant. The immunosuppressive effect of adenosine released by EVs requires further investigation, particularly in an ovalbumin mouse model. Using the ovalbumin mouse model may help us to study whether perforin can perforate any EVs. Puhka *et al.* reported that adenosine levels in prostate cancer (pre-prostatectomy) samples were lower than those in healthy controls or post-prostatectomy samples [41]. Considering their results alongside ours, there is a possibility that this phenomenon is reflected also in urinary EVs in prostate cancer patients. If adenosine from EVs has a profound or significant impact on the immune system, then adenosine-rich EVs may inhibit cytokine release syndrome, a side effect of CAR-T therapy. Furthermore, this study also suggests a novel type of EV-mediated intercellular communication mediated by perforin-induced EV destruction.

## Supporting information

**S1 Raw images.**
(PDF)

**S1 Table. Number of EV metabolites detected in each rate range.** From the metabolome profiling results for each EV, the proportion of each metabolite in the total metabolites of each EV was calculated, and the number of metabolites accounts for each proportion range was indicated. (DOCX)

**S1 Fig. Response of D3H2LN cells to IFN-γ.** To establish an experimental model to identify immunosuppressive metabolites, we first confirmed the responses of D3H2LN cells to IFN-γ. (A) Growth of D3H2LN cells in EV medium. D3H2LN cells ($1.0 \times 10^5$ cells/6 well) were seeded and cultured overnight in maintenance medium. After the medium was changed to EV medium, cell growth was measured at 0, 24, 48, and 72 h. (B) The transmission electron microscope image shows D3H2LN EVs 48 h after the medium was changed to EV medium. (C) Expression of indoleamine 2,3- dioxygenase 1 (IDO1) protein by D3H2LN cells cultured with or without IFN-γ for 24, 48, and 72 h. Bands on immunoblots were quantified using ImageJ software (http://rsb.info.nih.gov/nih-image/). (D) Number of D3H2LN cells. D3H2LN cells ($3 \times 10^6$ cells/150 mm$^2$) were seeded and cultured with or without IFN-γ. The number of cells was counted after 48 h. (E) Expression of indoleamine. 2,3-dioxygenase 1 (IDO1), programmed death-ligand 1 (PD-L1), CD9, CD63, Alix, Flotillin-1, TSG101, and apolipoprotein A1 (APOA1) in EVs from D3H2LN cells treated with or without IFN-γ. EV markers, CD9, CD63, Alix, Flotillin-1, and TSG101; EV negative protein marker, APOA1; immunosuppressive markers, IDO1 and PD-L1. (F) Average size of each EV, as assessed by NTA (left panel). As an example, size distribution of EVs derived from D3H2LN cells treated with or without IFN-γ for 48 h was assessed by NTA. All experiments were performed using three biological replicates.
(TIF)

**S2 Fig. Successive AFM images showing membrane disruption in IFN-γ EVs.** A solution of EVs from IFN-γ-treated D3H2LN cells (IFN-γ EVs) (14 ng in PBS) was incubated for 10 min on a mica surface in a humidified environment. Next, the mica was washed twice with PBS and imaged under an atomic force microscope (S4 Movie and S4 3D Movie: (A), S5 Movie and S5 3D Movie: (B)). USC-F1.2-k0.15 was used as a cantilever. B. Some of EVs from IFN-γ-treated D3H2LN cells (IFN-γ EVs) shrunk before bursting in response to perforin. The 3D images (processed using ImageJ software) derived from successive AFM images show a height reduction in IFN-γ EVs (red arrow). These images were obtained from the sample shown in Fig 2 but from a different area.
(TIF)

**S3 Fig. The average size of each EV from AFM data.** The diameters were calculated from cross-sectional image analysis of each AFM photo using ImageJ software. For this analysis, five vehicle EVs and eight IFN-γ EVs were analyzed.
(TIF)

**S4 Fig. Experimental scheme showing the EV membrane permeability assay.** Formation of perforin pores in the membrane of EVs from vehicle-treated D3H2LN cells (vehicle EVs) and EVs from IFN-γ-treated D3H2LN cells (IFN-γ EVs) was measured indirectly using real-time PCR to detect miR-16. Both types of EV were treated with perforin (0, 100, or 200 ng/mL) in the presence or absence of RNase A. After addition of an RNA lysis reagent, samples were spiked with cel-miR-39.
(TIF)

**S5 Fig. Scheme showing sample collection and metabolite extraction.** Cell samples, cell culture medium (CCM) samples, and EV samples for metabolomic profiling were collected according to this scheme. The EV medium was used as an indication of the background metabolic signature of the culture medium. UC, ultracentrifugation (110,000g for 70min at 4˚C); IS, internal standard.
(TIF)

**S6 Fig. Scheme showing metabolome analysis of EVs from IFN-γ-treated D3H2LN cells.** One-hundred-and-fifteen metabolites were detected in EV samples. Of these, 71 that were present in the EV samples at levels above twice the level detected in the blank were selected. Next, 30 metabolites present at consistently higher or lower levels in IFN-γ EVs were extracted. These metabolites were normalized in two ways: against particle number and against the amount of protein. Metabolites present in higher amounts in vehicle EVs than in IFN-γ EVs were detected by normalization against particles (n = 9) and by normalization against the amount of protein (n = 10). Of these, eight were detected after normalization against both particle number and amount of protein. These eight metabolites were not listed in the top ten metabolites detected according to absolute contribution rate values (Table 2). The total number of metabolites in IFN-γ EVs that exceeded those in vehicle EVs normalization against particles was 15, whereas the number after normalization against protein amount was 17; of these, 13 were detected by normalization against both particle number and protein amount. Of these 13 metabolites, five (uracil, uridine, adenosine, guanosine, and inosine) were listed in the top ten metabolites detected according to absolute contribution rate values (Table 2). All experiments were performed using three biological replicates.
(TIF)

**S7 Fig. Metabolome analysis of IFN-γ-treated D3H2LN cells, CCM, and EVs.** Cells and cell culture medium (CCM) samples were analyzed by CE-MS, and EV samples were analyzed by

IC-MS and LC-MS. Cells and CCM samples were also analyzed using LC-MS to detect uracil; this is because uracil is not detected accurately by CE-MS. A. Amount of uracil in cells, CCM, and EVs. B. Amount of uridine in cells, CCM, and EVs. C. Amount of guanosine in cells, CCM, and EVs. D. Amount of inosine in cells, CCM, and EVs. Columns, average concentration (cell samples; fmol/cell number ($10^6$); EV samples; fmol/mg protein); bars, S.D. (n = 3). N.D., the metabolite concentration was below the detection limit of the assay. $P < 0.05$ (Student's *t*-test). All experiments were performed using three biological replicates.
(TIF)

**S1 Movie. This movie was obtained using the same sample as in Fig 1A.** After 2.6 s, $CaCl_2$ was added to vehicle EVs. In this movie, EVs did not burst. The scan area is 1000 × 1000 nm. All images were taken at 1 frame per second (fps) and were edited at 10× speed (10 fps) using ImageJ.
(AVI)

**S2 Movie. This movie shows the same successive AFM images depicted in Fig 1A.** The movie starts 440 seconds after the perforin is added to the vehicle EVs. The EVs burst at 0:03 and 0:18. The scan area is 500 × 500 nm. All images were taken at 1 frame per second (fps) and were edited at 10× speed (10 fps) using ImageJ.
(AVI)

**S3 Movie. This movie shows the same successive AFM images depicted in Fig 1B.** The movie starts at 25 sec, before perforin was added to IFN-γ EVs. The EVs burst at 0:46. The scan area is 500 × 500 nm. All images were taken at 1 frame per second (fps) and were edited at 10× speed (10 fps) using ImageJ.
(AVI)

**S4 Movie. This movie shows the same successive AFM images depicted in S2A Fig.** The movie starts 1475 seconds after the perforin is added to the IFN-γ EVs. The EVs burst at 0:05. The scan area is 1000 × 1000 nm. All images were taken at 1 frame per second (fps) and were edited at 10× speed (10 fps) using ImageJ.
(AVI)

**S5 Movie. This movie shows the same successive AFM images depicted in S2B Fig.** The movie starts 1522 seconds after the perforin is added to the IFN-γ EVs. The EVs burst at 0:14. The scan area is 1000 × 1000 nm. All images were taken at 1 frame per second (fps) and were edited at 10× speed (10 fps) using ImageJ.
(AVI)

**S1 3D Movie. This 3D movie is a processed version of Supplementary Movie 1 (made using ImageJ).** 3D movies are created from 8-bit grayscale photos. The Z-axis of 255 is 70 nm. The 200.0 pixels in the X-axis and the Y-axis are 1000 nm. All 8-bit grayscale photos were taken at 10 fps.
(AVI)

**S2 3D Movie. This 3D movie shows the same 3D images depicted in Fig 1A.** The movie starts 440 seconds after the perforin is added to the vehicle EVs. The EVs burst at 0:03. 3D movies are created from 8-bit grayscale photos. The Z-axis of 255 is 50 nm. The 135.0 pixels in the X-axis is 337.5 nm. The 111.0 pixels in the Y-axis is 277.5 nm. All 8-bit grayscale photos were taken at 10 fps.
(AVI)

**S3 3D Movie. This 3D movie shows the same 3D images depicted in Fig 1B.** The movie starts at 25 sec, before perforin was added to IFN-γ EVs. The EVs burst at 0:46. 3D movies are

created from 8-bit grayscale photos. The Z-axis of 255 is 80 nm. The 119.0 pixels in the X-axis is 327.5 nm. The 119.0 pixels in the Y-axis is 297.5 nm. All 8-bit grayscale photos were taken at 10 fps.
(AVI)

**S4 3D Movie. This 3D movie shows the same 3D images depicted in S2A Fig.** The movie starts 1475 seconds after the perforin is added to the IFN-γ EVs. The EVs burst at 0:05. 3D movies are created from 8-bit grayscale photos. The Z-axis of 255 is 100 nm. The 105.0 pixels in the X-axis is 252 nm. The 112.0 pixels in the Y-axis is 560 nm. All 8-bit grayscale photos were taken at 10 fps.
(AVI)

**S5 3D Movie. This 3D movie shows the same 3D images depicted in S2B Fig.** The movie starts 1522 seconds after the perforin is added to the IFN-γ EVs. The EVs burst at 0:14. 3D movies are created from 8-bit grayscale photos. The Z-axis of 255 is 100 nm. The 54.0 pixels in the X-axis is 270 nm. The 46.0 pixels in the Y-axis is 225 nm. All 8-bit grayscale photos were taken at 10 fps.
(AVI)

## Acknowledgments

The authors thank Satsuki Ikeda (Institute for Advanced Biosciences, Keio University) for technical support with metabolomic profiling.

## Author Contributions

**Conceptualization:** Hiroko Tadokoro, Takahiro Ochiya.

**Data curation:** Akiyoshi Hirayama.

**Formal analysis:** Hiroko Tadokoro, Akiyoshi Hirayama.

**Funding acquisition:** Hiroko Tadokoro, Akiyoshi Hirayama.

**Investigation:** Hiroko Tadokoro, Masako Hasebe.

**Methodology:** Hiroko Tadokoro, Akiyoshi Hirayama, Ryuhei Kudo.

**Project administration:** Hiroko Tadokoro, Yusuke Yoshioka.

**Resources:** Hiroko Tadokoro, Akiyoshi Hirayama.

**Supervision:** Masahiro Sugimoto, Tomoyoshi Soga, Takahiro Ochiya.

**Validation:** Hiroko Tadokoro.

**Visualization:** Hiroko Tadokoro.

**Writing – original draft:** Hiroko Tadokoro.

**Writing – review & editing:** Juntaro Matsuzaki, Yusuke Yamamoto.

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
