## [Decision Letter · Decision Letter 0]

22 Jan 2020

PONE-D-19-35246

Self-sacrifice of cancer cell-derived extracellular vesicles Leakage of adenosine disrupts perforin-mediated burst of extracellular vesicles by cytotoxic T lymphocytes

PLOS ONE

Dear Dr. Ochiya,

Thank you for submitting your manuscript to PLOS ONE. After careful consideration, we feel that it has merit but does not fully meet PLOS ONE’s publication criteria as it currently stands. Therefore, we invite you to submit a revised version of the manuscript that addresses the points raised during the review process.

We would appreciate receiving your revised manuscript by Mar 07 2020 11:59PM. To enhance the reproducibility of your results, we recommend that if applicable you deposit your laboratory protocols in protocols.io, where a protocol can be assigned its own identifier (DOI) such that it can be cited independently in the future. For instructions see: http://journals.plos.org/plosone/s/submission-guidelines#loc-laboratory-protocols

We look forward to receiving your revised manuscript.

Kind regards,

Kazuhiro Takemoto

Academic Editor

PLOS ONE

Journal Requirements:

2. Please provide additional details regarding participant consent and IRB approval for sampling of human PBMCs. In the ethics statement in the Methods and online submission information, please ensure that you have specified (1) whether consent was suitably informed and (2) what type you obtained (for instance, written or verbal). If your study included minors under age 18, state whether you obtained consent from parents or guardians. If the need for consent was waived by the ethics committee, please include this information.

3. In your Methods section, please provide additional details regarding the cell lines used in your study and ensure you have described the source.

Additional Editor Comments:

The aim of the study is clear and the results are interesting. However, part of the manuscript is poorly constructed, as the reviewers mentioned. Thus, I require a major revision. However, I think the concerns the reviewers pointed out could be simply solved by adding more carefull explanations and by correcting the descriptions. I believe that the manuscript could meet the criteria for pubation after careful revision.

Reviewers' comments:

Reviewer's Responses to Questions

**Comments to the Author**

1. Is the manuscript technically sound, and do the data support the conclusions?

Reviewer #1: Yes

Reviewer #2: Partly

2. Has the statistical analysis been performed appropriately and rigorously? 

Reviewer #1: Yes

Reviewer #2: Yes

3. Have the authors made all data underlying the findings in their manuscript fully available?

Reviewer #1: Yes

Reviewer #2: Yes

4. Is the manuscript presented in an intelligible fashion and written in standard English?

Reviewer #1: No

Reviewer #2: Yes

5. Review Comments to the Author

Reviewer #1: The authors have attempted to elucidate the mechanisms employed by breast cancer cells derived EVs to overcome immune attack by CTLs. Here they show that perforin opens up EV membranes and results in adenosine release which can suppress CTL mediated perforin secretion. They also show the alleviation of this inhibition by the addition of ADA.

I believe that the authors need to address a few major points for the manuscript to be suitable for publication

1) Please edit your manuscript for sentence structure and some minor grammar. The text needs to be uniformly in past tense unless speculating or discussing your data.

2) Please include an explanation of why breast cancer cells were chosen for this study? Secondly, why did the authors limit their study to just one cell line. They could have easily included multiple breast cancer cell lines or alternately, looked at this phenomenon across tissue types

3) DO the authors exclude all serum (FBS) from their EV medium? Does that impact the cellular health as opposed to growing them in exosome depleted FBS?

4) Please edit the sub-section "sample collection and metabolite extraction:" within the methods section. It is unclear what the authors have processed in this section. Perhaps a schematic might be useful?

5) The movies included are very interesting, One suggestion would be to speed them up to 10X so that the viewers are not waiting for a long time for EV lysis.

6) Is it significant that the IFN EVs take longer to burst than the vehicle EVs? ( 7mins compared to 30 secs and 3 mins?

7) Consider combining both the panels in figures 2A and 2B into one. A stacked bar plot can express the data equally well and allow for comparisons between conditions

In the section "Metabolomic profiling of EVs"

8) Table 1 shows metabolites detected at >3%. However. The totals per group of EV is less than 70%. Does this mean that the remaining 30% were comprised of metabolites at very low concentrations? Please comment on this for both vehicle and IFN Evs

9) Per Table 2, all metabolites are increased or enriched after IFN addition( since all values are positive). Table 1 shows that at least for UDP-N_acetylglucosamine and ADP that it is not the case. How can we reconcile the contribution rates with table 1? Also shouldn’t some metabolites be decreased in IFN EVs when compared to vehicle EVs after being normalized to total metabolites (per table 2 caption)

10) Please edit line 319 where you reference Fig 3D (Fig 3 has not sub figures)

11) In lines 321 and 322: why are 3 values being reported? Please report average values and standard deviations

12) The authors have not characterized their EVs per MISEV 2018 guidelines. Please add the necessary characterizations

13) The authors refer to the measurement of pore sizes formed by perforin. Is it possible to measure the pore size induced in the EVs membranes by perforin? How do they compare to the sizes reported in literature( for cells)

14) The authors state that IFNG stimulation does dramatically change the metabolites in EVs. This could be true, but the authors need to allow for the fact that their stimulation concentration, times and cell type not be optimal for the effects to be observed

15) The authors have shown an interesting response of CTL's to tumor derived EV's but as they admit, this needs to be validated in a mouse model to ensure that it is a biologically relevant mechanism. Additionally, as they point out, only 15% of the perforin secretion is reduced. Is the remaining 85% perforin secretion sufficient to cause lysis?

16) I assumed that the source of adenosine in the EVs is from the cells, but from the discussion it seems like cellular adenosine is short lived and rapidly converted to inosine. How is adenosine incorporated into Evs then?

17) The authors should consider using multiple miRNA instead of just miR 16 for figure 2 in order to overcome the limitations of relying on a single miRNA to estimate EV integrity.

Reviewer #2: The manuscript “Self-sacrifice of cancer cell-derived extracellular vesicles leakage of adenosine disrupts perforin-mediated burst of extracellular vesicles by cytotoxic T lymphocytes” by Tadokoro et al., is a fluent and interesting read. As the authors address the young field of extracellular vesicles, the minuscule lipid membrane bound carriers of molecules, they are challenged by the uncertainties related to the success of purification and close-to-the detection limit data. However, the challenges are generally very well discussed, and therefore I feel that tackling some remaining issues would help the authors to provide a more comprehensive publication . Here are my specific comments:

-Title is too long and difficult to understand. In addition, the use of “self-sacrifice” sounds odd, when it describes the activity of EVs. Thus remove self-sacrifice and shorten the title.

-The data shows that perforin makes holes to some cancer-derived EVs (+/-IFNy treatment), which then leak adenosine. To provide more convincing data and knowledge of the extent of this phenomenon, the amount (e.g. the %) of bursting/shrinking EVs should be quantified. In addition, as a control, it would be good to show, whether perforin can perforate any EVs (fibroblast EVs etc +/-IFNy treatment) and whether the control EVs leak adenosine/reduce perforin secretion by CTLs as well.

-Related to the previous, I found the miR assay results shown in Fig. 2A somewhat difficult to understand. Even if no RNAse is added, I would have expected that some natural RNAses remain in the EV preparations, that would degrade the miR-16 released from EVs when perforin is added. Instead, the quantification shows no degradation after addition of perforin. To support the AFM data about EV disruption, perhaps the best option would be to add some other data, for example to perform super-resolution imaging of EVs labeled with a soluble dye (+/- perforin).

-Matrials and methods: section about western blotting lacks some basic information, such as the blot type, transfer system and ECL kit used.

-Fig.3 legend text tells that tryptophan is increased, but the main text and the figure bar graph show that it is decreased. This should be corrected.

-Overall, the discussion is well written and addresses several different factors in the tumor microenvironment. The discussion also refers to an earlier publication (ref. 40) showing enrichment of adenosine in urine and platelet EVs. The same publication also contained a preliminary study of different normalization methods and metabolite changes of urinary EVs in prostate cancer. There, adenosine was found to be decreased in cancer (pre-prostatectomy) samples compared to healthy controls or post-prostatectomy samples. In the light of the results about perforin-mediated lysis of EVs and release of adenosine in the tumor microenvironment, there is a possibility that this phenomenon is reflected also in the urinary EVs in prostate cancer thus supporting the conclusions by Tadokoro et al.

6. PLOS authors have the option to publish the peer review history of their article (what does this mean?). If published, this will include your full peer review and any attached files.

Reviewer #1: No

Reviewer #2: No

---

## [Author Response · Author response to Decision Letter 0]

7 Mar 2020

Please see the attached file "Response_to_Reviewers_Tadokoro_PONE-D-19-35246.docx".

---

## [Decision Letter · Decision Letter 1]

24 Mar 2020

Adenosine leakage from perforin-burst extracellular vesicles inhibits perforin secretion by cytotoxic T-lymphocytes

PONE-D-19-35246R1

Dear Dr. Ochiya,

We are pleased to inform you that your manuscript has been judged scientifically suitable for publication and will be formally accepted for publication once it complies with all outstanding technical requirements.

With kind regards,

Kazuhiro Takemoto

Academic Editor

PLOS ONE

Additional Editor Comments (optional):

The authors carefully revised the manuscript. I believe that the manuscript is suitable for publication. However, to increase readability, the authors can improve the manuscript according to the comments of Reviewer 2. I recommend the authors to modify the manuscript.

Reviewers' comments:

Reviewer's Responses to Questions

**Comments to the Author**

1. If the authors have adequately addressed your comments raised in a previous round of review and you feel that this manuscript is now acceptable for publication, you may indicate that here to bypass the “Comments to the Author” section, enter your conflict of interest statement in the “Confidential to Editor” section, and submit your "Accept" recommendation.

Reviewer #1: All comments have been addressed

Reviewer #2: (No Response)

2. Is the manuscript technically sound, and do the data support the conclusions?

Reviewer #1: Yes

Reviewer #2: Yes

3. Has the statistical analysis been performed appropriately and rigorously? 

Reviewer #1: Yes

Reviewer #2: I Don't Know

4. Have the authors made all data underlying the findings in their manuscript fully available?

Reviewer #1: Yes

Reviewer #2: Yes

5. Is the manuscript presented in an intelligible fashion and written in standard English?

Reviewer #1: Yes

Reviewer #2: Yes

6. Review Comments to the Author

Reviewer #1: I believe that the authors have addressed all my comments satisfactorily and that the manuscript is now suitable for publication

Reviewer #2: While I think that the authors have reponded to all the major points raised, few comments remain. Statistical section should provide information, whether and how normality of the data was analyzed. It would also improve the view on the statistics in the publication, if the authors added a clear statement, how many EVs they observed altogether and how many of those were witnessed to burst.

7. PLOS authors have the option to publish the peer review history of their article (what does this mean?). If published, this will include your full peer review and any attached files.

Reviewer #1: No

Reviewer #2: No

---

## [Editor Report · Acceptance letter]

30 Mar 2020

PONE-D-19-35246R1 

Adenosine leakage from perforin-burst extracellular vesicles inhibits perforin secretion by cytotoxic T-lymphocytes 

Dear Dr. Ochiya:

I am pleased to inform you that your manuscript has been deemed suitable for publication in PLOS ONE. Congratulations! Your manuscript is now with our production department. 

With kind regards,

on behalf of

Dr. Kazuhiro Takemoto 

Academic Editor

PLOS ONE